# Real-World Gestational Diabetes Screening: Problems with the Oral Glucose Tolerance Test in Rural and Remote Australia

**DOI:** 10.3390/ijerph16224488

**Published:** 2019-11-14

**Authors:** Emma L. Jamieson, Erica P. Spry, Andrew B. Kirke, David N. Atkinson, Julia V. Marley

**Affiliations:** 1The Rural Clinical School of Western Australia, The University of Western Australia, Bunbury 6233, Australia; andrew.kirke@rcswa.edu.au; 2Kimberley Aboriginal Medical Services, Broome 6725, Australia; research@kamsc.org.au (E.P.S.); julia.marley@rcswa.edu.au (J.V.M.); 3The Rural Clinical School of Western Australia, The University of Western Australia, Broome 6725, Australia; david.atkinson@rcswa.edu.au

**Keywords:** gestational diabetes, GDM, oral glucose tolerance test, OGTT, glycolysis

## Abstract

Gestational diabetes mellitus (GDM) is the most common antenatal complication in Australia. All pregnant women are recommended for screening by 75 g oral glucose tolerance test (OGTT). As part of a study to improve screening, 694 women from 27 regional, rural and remote clinics were recruited from 2015–2018 into the Optimisation of Rural Clinical and Haematological Indicators for Diabetes in pregnancy (ORCHID) study. Most routine OGTT samples were analysed more than four hours post fasting collection (median 5.0 h, range 2.3 to 124 h), potentially reducing glucose levels due to glycolysis. In 2019, to assess pre-analytical plasma glucose (PG) instability over time, we evaluated alternative sample handling protocols in a sample of participants. Four extra samples were collected alongside routine room temperature (RT) fluoride-oxalate samples (FLOX^RT^): study FLOX^RT^; ice slurry (FLOX^ICE^); RT fluoride-citrate-EDTA (FC Mix), and RT lithium-heparin plasma separation tubes (PST). Time course glucose measurements were then used to estimate glycolysis from ORCHID participants who completed routine OGTT after 24 weeks gestation (*n* = 501). Adjusting for glycolysis using FLOX^ICE^ measurements estimated 62% under-diagnosis of GDM (FLOX^RT^ 10.8% v FLOX^ICE^ 28.5% (95% CI, 20.8–29.5%), *p* < 0.001). FC Mix tubes provided excellent glucose stability but gave slightly higher results (Fasting PG: +0.20 ± 0.05 mmol/L). While providing a realistic alternative to the impractical FLOX^ICE^ protocol, direct substitution of FC Mix tubes in clinical practice may require revision of GDM diagnostic thresholds.

## 1. Introduction

Due to the high global incidence of gestational diabetes mellitus (GDM) and potential for improved perinatal outcome, the International Association of Diabetes in Pregnancy Study Groups (IADPSG) recommends all pregnant women, without pre-existing diabetes, be screened for hyperglycaemia by 75 g oral glucose tolerance test (OGTT) [1,2]. In 2008, the IADPSG convened a consensus panel of member organisations to review several studies that demonstrated a continuous linear association between mild maternal hyperglycaemia, measured by OGTT, and adverse perinatal outcomes [3,4,5,6,7]. In 2010, the IADPSG consensus panel derived new OGTT diagnostic cut-points for GDM based only on Hyperglycaemia and Adverse Pregnancy Outcomes (HAPO) study data.

HAPO data was selected due the robust study design, including strict pre-analytical OGTT protocol, which was standardised across multiple international sites [2,3]. Fluoride has historically been used to stabilise glucose in OGTT samples; however, glycolysis is not inhibited over the first hour post-collection and takes four hours to completely inhibit glycolysis [8]. The World Health Organization (WHO) first recommended rapid plasma separation from fluoride collections in their *Diabetes Mellitus* 1985 study group report, in acknowledgement of delays in glycolytic inhibition [9]. To minimise glycolysis, the HAPO study pre-analytical OGTT protocol stipulated immediate immersion of fluoride-oxalate (FLOX) samples in ice-slurry or crushed-ice, prior to plasma separation [10,11]. This is impractical in most routine clinical and laboratory settings and has not been implemented in Australia; most laboratories batch transport FLOX samples without recommendation for storage on ice or for rapid plasma separation (Table A1).

Other countries use pragmatic alternatives to the HAPO protocol to address pre-analytical glucose instability. The American Diabetes Association (ADA) national guidelines recommend collection into tubes with an immediate glycolytic inhibitor, such as citrate, if delays of more than 30 min are anticipated [12]. Reiterated in diabetes guidelines across Europe, 2008 European Standards recommend combined fluoride-citrate tubes for immediate and long-term stabilisation of plasma glucose [13,14,15,16]. Rapid centrifugation of lithium-heparin plasma separation tubes (PST) is considered a gold-standard method to minimise glycolysis for research purposes [17].

The impact of glycolysis on GDM diagnosis can be significant. Daly et al. observed a 2.7-fold increase in GDM by HAPO protocol, compared to routine hospital procedures [18]. The group suggested GDM under-diagnosis was further compounded by longer delays to analysis for OGTT collected at external clinics (GDM incidence: 17.8% hospital clinic v 10.6% external clinic, 30 km from laboratory) [19]. Using a basic linear adjustment of retrospective data from a large Victorian regional centre, Song et al. also estimated that there would be a substantial increase in GDM if fluoride-citrate-EDTA (FC Mix) tubes were used instead of routine procedures (39.2% vs. 13.5% by standard procedures) [20]. However, they suggested that glycolysis had no clinical impact as they reported the same proportion of large-for-gestational-age (LGA) babies in the lower HAPO glucose categories.

Large distances and delays to laboratory analysis are common in rural and remote Australia. The prospective Optimisation of Rural Clinical and Haematological Indicators for Diabetes in pregnancy (ORCHID) study was conducted in regional, rural and remote Western Australian clinical settings. Only 50% of women complete screening for GDM by OGTT in these settings [21]. ORCHID aimed to investigate ways to improve screening for GDM and relied on OGTT collected using local clinical and laboratory protocols. As part of ORCHID we explored the potential impact of glycolysis and a pre-analytical protocol change on GDM diagnosis. This paper aims to describe a detailed analysis of glucose decline over time for fasting, 1-h and 2-h OGTT samples in FLOX tubes stored at room temperature. It also aims to model the potential impact of estimated glycolysis on GDM incidence in the ORCHID study cohort.

## 2. Materials and Methods

Setting: In Western Australia (WA), 21.3% of pregnant women reside regionally [22]. Maternity services are facilitated by public hospital maternity care, midwifery group practice care, community midwifery/shared care, general practitioner obstetric care or team midwifery care. Sample selection for the ORCHID study was biased to over-represent Aboriginal women to allow sub-cohort analysis of this high-risk group.

Data were collected by local health care providers from 9 January 2015 to 31 May 2018, at 27 sites in the Kimberley, Mid-West, Goldfields, Southwest and Great Southern regions of WA. The Modified Monash Model (MMM) was used to classify clinic location; for binary analysis remote clinics (MMM6 and MMM7) were classified as MMM ≥ 6.

Recruitment and data collection of the ORCHID cohort: Pregnant women presenting for their first antenatal visit at a participating site, aged 16 years or older, no documented pre-existing diabetes, and singleton pregnancy, were invited to take part. Maternal characteristics, including known risk factors for GDM, were recorded by antenatal care providers at recruitment and at time of routine OGTT after 24 weeks gestation. Birth outcomes were obtained from hospital discharge summaries and special care nursery discharge summaries.

Prospective ORCHID cohort OGTT and glucose measurement: Local clinical and laboratory procedures were relied on for 75 g OGTT collection, transport and laboratory analysis. Participants were classified as having GDM if one or more IADPSG 2010 glucose criteria were met [2]:Fasting plasma glucose ≥ 5.1 mmol/L1-h plasma glucose ≥ 10.0 mmol/L2-h plasma glucose ≥ 8.5 mmol/L

A survey of all ORCHID study sites revealed that local pathology specimen collection protocols did not use the same pre-analytical protocol as HAPO. All venous whole blood samples were collected into FLOX tubes (BD Biosciences, Australia) and batch sent to local pathology laboratories. Sample storage and transportation temperature varied by site (refrigerated or room temperature). Distances between study sites and laboratory ranged from 0.35 km by road to 650 km by air. Upon arrival at the pathology laboratory, plasma was separated for measurement of glucose by hexokinase method [23]. Time delay between collection and analysis was recorded for 424 OGTT; minimum delay by site was used when this data was missing (*n* = 77). A separate sub-cohort of seven participants from a remote clinic had paired FC Mix (Greiner Bio-One, Austria) collections alongside standard OGTT.

Recruitment of the OGTT time-course group: Pregnant women presenting for their routine 24–28 week OGTT at a metropolitan hospital, aged 16 years or older, no documented pre-existing diabetes and singleton pregnancy, were invited to take part.

Time-course OGTT and adjustment algorithms: To generate algorithms to adjust ORCHID cohort OGTT results for estimated glycolysis due to delay to analysis, 12 pregnant women in a metropolitan hospital setting had additional samples collected alongside their routine OGTT from 24 May 2019 to 5 July 2019 (Figure 1). Eleven women completed OGTT. Complete tube fill was ensured for all samples:FLOX^RT^ protocol: FLOX tube (5 mL BD Biosciences, Australia) stored at room temperature. Aliquots collected at 0.25 h, 0.5 h, 0.75 h, 1 h, 1.5 h, 2 h, 2.5 h, 3 h, 3.5 h, and 4 h.FLOX^ICE^ protocol: FLOX tube (2 mL BD Biosciences, Australia) placed immediately in ice-slurry. Aliquots collected at 0.25 h, 0.5 h, 0.75 h, and 1 h.FC Mix protocol: FC Mix tube (3 mL) stored at room temperature. Aliquots collected at 0.25 h, 0.5 h, 0.75 h, 1 h, and 4 h.PST protocol: PST (3 mL BD Biosciences, Australia), centrifuged within 5 min of collection at 1300× *g* for 5 min, and stored at room temperature. Separated plasma fraction remained in the collection tube for the entire experiment.

Following four hours at room temperature, FLOX^RT^, FC Mix, and PST tubes from six participants were also refrigerated overnight for 24 h plasma glucose analysis. All whole blood aliquots (300 μL) were centrifuged at 1300× *g* for 5 min and 100 μL of plasma removed for plasma glucose analysis.

Plasma glucose aliquots were immediately analysed by glucose oxidase, hydrogen peroxide method (Ortho-Clinical Diagnostics, Vitros 5600 Integrated System; median time to completion of analysis was 8.2 min, IQR 6.2 to 10.2 min). Inter-assay CV was < 2.17% and haemolysis, icterus, and turbidity index for all samples were below local interference thresholds. Upon arrival in the same laboratory, plasma from the standard OGTT samples was separated for measurement of glucose by the same method.

Baseline FLOX^ICE^ glucose was calculated as mean FLOX^ICE^ glucose from plasma separated < 1 h post-collection based on ice stabilisation assessment (Table A2). Baseline FC Mix glucose was calculated as mean FC Mix glucose from plasma separated ≤ 4 h. Beta coefficients were calculated by linear regression for plasma glucose, delay and delay-squared to estimate baseline FLOX^ICE^ or baseline FC Mix (Table A3). A separate estimate for GDM by FC Mix protocol was calculated for participants within MMM remoteness categories 2 and 3 using the basic linear algorithm used by Song et al.: plasma glucose multiplied by 1.113 for fasting samples; 1.081 for 1-h samples and 1.061 for 2-h samples (Table A4) [20]. IADPSG 2010 criteria were applied to algorithm-adjusted ORCHID cohort OGTT results to obtain adjusted GDM incidence.

Statistical analysis: Study data were collected and managed using secure REDCap electronic data capture tools hosted at The University of Western Australia [24]. All analyses were performed with Stata, version 14 (Statacorp, College Station, United States of America). For LGA infants (birth weight greater than 90th centile), birth weight centiles were calculated using the Global bulk centile calculator, GROW v8.0.1 (Perinatal Institute, Birmingham, United Kingdom), adjusting for gestational age, maternal height, maternal weight at first antenatal visit, parity, ethnicity, and infant sex. Adjustment for maternal weight was made within body mass index (BMI, maternal weight/height^2^ (kg/m^2^)) limits of 18.5–30 kg/m^2^ only. Differences in characteristics between groups were compared using *t*-tests for continuous data and χ^2^ tests for categorical data, with the exception of differences in remoteness classification, which was compared by Cuzick non-parametric test for trend across ordered groups. Differences in glucose concentrations between paired OGTT samples were compared using *t*-tests. Differences between standard and adjusted GDM incidence were compared using McNemar’s test. Error for adjusted GDM incidence for FLOX^ICE^ and FC Mix algorithms was calculated from proportional 95% CI obtained from validation of algorithms in time-course participants. Low and high 95% CI values were used to adjust IADPSG fasting, 1-h and 2-h plasma glucose cut-points to obtain a 95% CI for adjusted GDM incidence. A *p*-value of less than 0.05 was defined as statistically significant.

Ethics approval: Ethics approval was obtained from the Western Australian Aboriginal Health Ethics Committee (2014-007) and WA Country Health Service Research Ethics Committee (2014-16). The Kimberley Aboriginal Health Planning Forum Research Subcommittee supported this project.

## 3. Results

Six hundred and ninety-four women were enrolled in ORCHID. Six hundred ORCHID participants delivered their babies after 30 weeks gestation. This represented 2.7% of 22,513 births to rural WA women over the period of the ORCHID Study (WA Midwives Notification System, data extract date: 4 December 2018).

Eleven ORCHID participants with GDM detected by OGTT between 6- and 12-weeks gestation were excluded from analysis. Of the remaining 589 ORCHID participants recommended for OGTT completion after 24 weeks gestation, only 85.1% completed an OGTT despite amenability at recruitment (Figure 1). Participants of high-risk Aboriginal or Torres Strait Islander ethnicity, hereafter referred to as Aboriginal, were younger (26.0 ± 5.2 years vs. other ethnicity 30.2 ± 5.3 years, *p* < 0.001) more likely to have a family history of type 2 diabetes (98/229, 42.8% vs. other ethnicity 83/360, 23.1%, *p* < 0.001) and more likely to decline or not complete the OGTT (Table 1). Aboriginal women who did complete the OGTT, did so at a later gestational age (28.8 ± 2.4 weeks vs. other ethnicity 27.3 ± 1.5 weeks, *p* < 0.001). Aboriginal women also commenced OGTT later in the morning (median (IQR) time of fasting plasma glucose collection: Aboriginal 9:17 a.m. (8:48 a.m. to 9:45 a.m.) vs. other ethnicity; 8:35 a.m. (8:10 a.m. to 9:10 a.m.), *p* < 0.001).

GDM incidence after 24 weeks gestation was lower than expected in this high-risk cohort (GDM: 10.8%; 54/501 complete OGTT). Almost half of ORCHID fasting plasma glucose samples fell within the lowest of seven glucose categories defined in HAPO (2.5–4.1 mmol/L: 44.9%, 225/501 vs. HAPO 17.4%, 4043/23,231, *p* < 0.001). After exclusion of 49 participants who were managed for GDM, incidence of LGA reached 10% at HAPO fasting glucose category one (10.7%, 23/192); 1-h glucose category two (5.9–7.3 mmol/L: 10.0%, 17/170) and 2-h glucose category two (5.1–6.0 mmol/L: 13.2% 21/159). Most ORCHID OGTT samples were stored at room temperature (92%, 461/501) and had long delay to laboratory analysis (median time to analysis for fasting plasma glucose: 5.0 h; range 2.3 to 124 h).

In the time-course analysis, plasma glucose measurements were significantly lower in FLOX^RT^ samples held at room temperature for four hours, compared to baseline FLOX^ICE^ protocol (mean difference ± SD (mmol/L): fasting plasma glucose: −0.43 ± 0.07, *p* < 0.001; 1-h plasma glucose: −0.48 ± 0.17, *p* < 0.001; 2-h plasma glucose: −0.44 ± 0.19, *p* < 0.001, Figure 2).

PST, centrifuged immediately, gave similar glucose values to baseline plasma glucose by FLOX^ICE^ protocol (Figure 2). However, significant loss of glucose was observed in PST samples at 24 h (−0.22 mmol/L ± 0.16 vs. PST at 5 min, *p* < 0.001). Only FC Mix tubes provided 24 h plasma glucose stability (mean difference ± SD (mmol/L) at 24 h compared to baseline FC Mix measurement: fasting plasma glucose, −0.03 ± 0.07, *p* = 0.398; 1-h plasma glucose, −0.02 ± 0.09, *p* = 0.674; 2-h plasma glucose, −0.02 ± 0.05, *p* = 0.427, Figure 2). Notably, 25.6% (34/133) of OGTT from remote ORCHID study sites were not analysed on the day of collection.

Greater variability in glycolysis between individuals was observed in the 1-h and 2-h OGTT samples, compared to fasting. Despite this variability, the FLOX^ICE^ and FC Mix algorithms showed high validity across all three OGTT time-points when used to adjust standard OGTT results from time-course participants (Table 2 and Table 3). Minimal variation in delay to analysis from fasting collection was observed in this metropolitan hospital setting (median delay 3.1 h, range 2.4–3.7 h). In the other paired-sample analysis conducted at a remote clinic, adjusted standard OGTT glucose concentrations were comparable to those obtained from FC Mix tubes; however, a slight and insignificant overestimate of glycolysis at the 2-h sample was observed (mean (95% CI: absolute (mmol/L) and proportional (%)) difference in adjusted plasma glucose compared to FC Mix plasma glucose: fasting plasma glucose, 0.04 ((−0.01 to 0.09) −0.1% to 2.0%), *p* = 0.13; 1-h plasma glucose, 0.05 ((−0.06 to 0.17), −0.69% to 2.34%), *p* = 0.37; 2-h plasma glucose, 0.17 ((−0.03 to 0.37) −0.9% to 5.5%), *p* = 0.14).

Adjustment to account for reduced glycolysis under HAPO study (FLOX^ICE^) conditions revealed significant under-diagnosis of GDM in the ORCHID cohort due to standard pre-analytical protocols (GDM: standard FLOX, 10.8% vs. FLOX^ICE^ 28.5% (95% CI, 20.8–29.9%, *p* < 0.001). FC Mix protocol substantially increased GDM incidence to 45.3% (95% CI, 35.7–55.1%) of the cohort screened (Table 4). Increases in GDM incidence by both FLOX^ICE^ and FC Mix protocol were predominantly due to increased detection of hyperglycaemia in fasting OGTT samples (FLOX^ICE^ estimate: 81.1% with GDM diagnosed by fasting plasma glucose; FC Mix estimate: 86.3% with GDM diagnosed by fasting plasma glucose). Application of the linear algorithm for ORCHID participants whose OGTT was collected in larger regional sites underestimated GDM incidence compared to our non-linear adjustment (32.1% by linear algorithm vs. 50.5% by FC Mix algorithm, *p* < 0.001, Table A4).

Fasting plasma glucose was significantly lower in Aboriginal participants compared to the rest of the ORCHID cohort, even after adjustment. Consequently, the proportion of Aboriginal participants estimated to have GDM by adjusted fasting plasma glucose alone was also lower, as was overall estimated GDM incidence by FC Mix protocol (Table 4).

## 4. Discussion

We estimate 62% of regional, rural and remote Western Australian women with GDM are not diagnosed by 24–28 week OGTT due to a sample processing issue. Whilst plasma glucose instability has been raised as an important consideration by some groups [18,25], the extent to which glycolysis impacts GDM diagnosis for women in diverse rural and remote Australian settings has not been previously characterised. As observed in other cohorts, glycolysis had the largest impact on GDM diagnosis by fasting plasma glucose [18,26]. This is unsurprising considering batched fasting OGTT samples have the longest delay to analysis and narrowest standard deviation: 0.5 mmol/L in ORCHID and 0.4 mmol/L in HAPO. This rendered the small average HAPO FLOX^ICE^ adjustment of 0.44 mmol/L ± 0.1 in fasting glucose highly significant, and was 3.7 times larger than seasonal variation reported in the Brisbane and Newcastle HAPO study sites (+0.12 mmol/L) [27].

However, the HAPO FLOX^ICE^ protocol is impractical in most settings. If we are to acknowledge the impact glycolysis has on identification of women for GDM management, suitable alternate protocols to stabilise glucose must be explored. Plasma glucose from PST closely matched HAPO FLOX^ICE^ values over four hours, however the PST protocol has several disadvantages requiring consideration: necessity for immediate access to a centrifuge in collection area; increase in sample-handling burden on clinic staff (in our setting this would predominately be midwives), and potential for glycolysis to continue unabated if tubes are not centrifuged immediately and for sufficient time. In addition, poor long-term stability of glucose, also observed in other stability assessments [28] would render the PST protocol an unsatisfactory alternative for remote locations, where 25.6% were not processed on the same day they were collected.

In contrast, fluoride-citrate-EDTA (FC Mix) tubes ensured both short and long-term glucose stability and substitution of FC Mix tubes into clinical practice would not require additional changes to clinic collection protocols. However, revision of diagnostic criteria may be warranted as FC Mix plasma glucose values were, on average, 0.2 mmol/L higher than those obtained by research grade methodology. Reports of citrate or fluoride-citrate glycolytic inhibition compared to research grade methodology in the literature are inconsistent, ranging from no observed difference in plasma glucose to +0.34 mmol/L higher (reviewed by Carey et al.) [29]. These differences are potentially related to method variations for participant selection, research grade anticoagulant/inhibitor, cooling, timing of plasma separation or use of liquid versus lyophilised citrate buffer. For direct comparison of GDM by IADPSG criteria in our study, we recruited pregnant women at routine OGTT, strictly followed the HAPO study protocol and selected lyophilised FC Mix tubes in order to avoid potential issues with dilution factor corrections. The small but significantly higher glucose estimates by FC Mix protocol had the largest impact on GDM diagnoses by adjusted fasting plasma glucose and resulted in 1.8-fold higher estimation of GDM compared to FLOX^ICE^ protocol.

As Song et al. used retrospective data, the delay to analysis was unknown; the authors applied a basic linear adjustment, assuming a uniform delay of 3.6 h from fasting collection, 2.4 h from 1-h collection and 1.6 h from 2-h collection [20]. However, delays to analysis in our cohort were larger and varied between sites, even within geographically similar sites to the Victorian cohort, explaining the underestimation of GDM by linear algorithm compared to our non-linear adjustment. The authors also speculated that the FLOX^ICE^ protocol was likely not adhered to by HAPO study sites, using poor adherence to time restrictions for processing cord-blood samples as an indication. They suggested that a protocol change to minimise glycolysis is not warranted based on the lack of a shift towards higher rates of LGA babies in lower HAPO glucose categories in their cohort. By contrast, we observed rates of LGA above 10% at the two lowest glucose categories. We also showed that incubation on ice stabilised glucose concentrations in FLOX tubes by 1 h post-collection, therefore it is unlikely that further delay to analysis for HAPO study OGTT samples would have lowered glucose measurements (Table A2). The Victorian cohort represented a low-risk, predominantly Caucasian population, with significantly lower BMI and birth weight outcomes compared to the ORCHID cohort, which may explain some of the differences in LGA incidence distribution between cohorts (BMI: Song et al. 27.7 ± 0.20 vs. ORCHID 28.5 ± 0.27, *p* = 0.026; birth weight: Song et al. 3376g ± 502 vs. 3458 ± 531, *p* = 0.003) [20]. The authors also reported that plasma was separated in all OGTT samples prior to transportation to laboratory. This would have reduced glycolysis significantly; however, this is not the procedure used in Western Australia and would be difficult to implement where OGTTs are not conducted in laboratory settings.

Increases in GDM by FLOX^ICE^ or FC Mix protocol appeared attenuated in Aboriginal women, due to lower fasting glucose compared to the rest of the ORCHID cohort. This difference may be due to sampling bias; Aboriginal women were younger and more likely to decline OGTT and may therefore not be representative of the underlying population. Another explanation could be increased foetal glucose steal activity, as Aboriginal women tended to complete OGTT later in gestation. Aboriginal participants also tended to commence OGTT later in the morning. Established diurnal trends for declining fasting glucose throughout the morning, even after adjustment for fasting duration, may have contributed to the lower fasting glucose observed in Aboriginal participants [30,31]. Therefore, whilst our estimates suggest the impact of implementation of FC Mix tubes on GDM diagnosis may not be as great in Aboriginal women who do complete an OGTT, caution is advised when interpreting differences in this ethnic group.

It is a study limitation that our GDM estimates for the ORCHID cohort are based on algorithmic adjustment; however, a direct comparison of standard and alternate OGTT protocols to HAPO FLOX^ICE^ protocol was not feasible in our rural and remote setting. We acknowledge that the 1-h and 2-h algorithms were not as robust as the fasting, due to greater inter-individual variation of glycolysis in the post-load samples. This is an additional source of imprecision, and a simple, standard adjustment of routine OGTT results to correct for glycolysis would likely add to measurement uncertainty. However, as the fasting algorithms were robust, and the pattern of predominant diagnosis by fasting sample held true for both FLOX^ICE^ and FC Mix adjustments, it is likely our FC Mix GDM estimates will prove to be accurate when tested prospectively. Furthermore, by standardising pre-analytical processes for glucose, screening by fasting plasma glucose alone, as suggested by other groups, may become a viable option to improve screening coverage in our population with poor OGTT completion [21,32].

The HAPO study did not document time to plasma separation. As FLOX^ICE^ plasma glucose was stabilised by 45 min, we calculated mean plasma glucose from FLOX^ICE^ protocol 15 min, 30 min, and 45 min samples as an estimate of plasma glucose by HAPO protocol. However, other groups have demonstrated that the largest drop in glucose following ice-immersion occurs within the first 15 min post-collection [33,34]. Despite attaining core tube temperatures of 5 °C at 3 min post-ice-immersion, the reported inter-individual glycolytic rate at 15 min is highly variable (range: no change to −0.29 mmol/L, *n* = 7). Our study did not include a time-0 FLOX measurement. This may have biased baseline FLOX^ICE^ calculations towards lower glucose concentrations compared to HAPO, had immediate centrifugation been achieved.

Due to the geographical diversity of participating sites, it is likely that temperatures prior to and during sample transportation varied. This information, which could have a significant impact on glycolysis, is unknown. In addition, the error for GDM estimates by adjusted OGTT is calculated from time-course participant data. As our adjustment algorithms were also based on time-course data, it is possible that this error is underestimated. When the FC Mix algorithm was applied to the remote cohort with paired FC Mix tube collections, algorithmic adjustment was biased towards a small over-estimation of glucose at the 2-h sample.

Another study limitation is that the error around adjusted GDM incidence does not include analytical error. OGTT samples were measured at local laboratories state-wide, with pathology provider based on clinician choice. All laboratories were National Association of Testing Authorities, Australia (NATA) accredited, however the magnitude of imprecision and inter-laboratory bias for glucose measurements at each laboratory is unknown. ADA 2011 recommendations for glucose measurement are for imprecision ≤ 2.9%, and bias ≤ 2.2%, allowing a total analytical error (TAE) of ≤ 6.9% [12]. TAE for glucose, even within allowable limits, can have a significant impact on GDM incidence. Agarwal et al. applied the allowable TAE to IADPSG criteria to obtain a 95% confidence interval for each diagnostic threshold: fasting plasma glucose, 4.7–5.5 mmol/L; 1-h plasma glucose, 9.3–10.7 mmol/L; and 2-h plasma glucose, 7.9–9.1 mmol/L [35]. This translated to a 0.5- to 1.7-fold change in GDM incidence in a United Arab Emirates cohort, depending on threshold used.

For appropriate interpretation of diagnostic results, clinicians must be aware of measurement uncertainty, including biological, pre-analytical, and laboratory variation. However, it is not a requirement to report estimates of measurement uncertainty with pathology results. The Royal College of Pathologists of Australasia and the Australasian Association of Clinical Biochemists encourage clinicians to contact their pathology provider to ascertain measurement uncertainty for each assay [36,37]. Laboratories will only report TAE, without acknowledgment of sources of pre-analytical error. For samples at the diagnostic fasting threshold for GDM, the 4 h drop in glucose due to glycolysis alone exceeds the allowable TAE.

Whilst site and laboratory differences could be considered a study limitation, they also highlight the difficulties of consistent implementation of research protocols in real world settings and strengthen the argument for pre-analytical standardisation. It should also be noted that centralisation of laboratory measurement for research is not without problems. Several months of OGTT data from HAPO were algorithmically adjusted following discovery of bias between the central laboratory and all other site laboratories [10]. The bias was related to an inconsistent batch of analytical reagents.

The introduction of the IADPSG criteria raised concerns about the likelihood of overwhelming numbers of women diagnosed with GDM and the burden on antenatal care services that would follow. Figures from this study and others suggest this has not happened. We believe, in our cohort of rural women, a significant number who should be recognised as having GDM by the IADPSG criteria are being missed due to processing issues. The biggest impact of addressing this issue will be felt on the women diagnosed on fasting blood glucose. A significant proportion of these will be managed with diet, exercise, and timed delivery at the end of pregnancy. A smaller proportion will require greater clinical input, insulin, or other hypoglycaemic medication and more specialist intervention. Overall the major impact of standardising pre-analytical processes is likely to be the need for a more comprehensive education program about blood sugar control for all pregnant women, as greater numbers will be picked up in the lower risk end of the spectrum. The true impact of the IADPSG 2010 criteria will also become visible with the removal of the effect of pre-analytical glycolysis in GDM screening.

## 5. Conclusions

This study evaluated the impact of glycolysis on GDM incidence in a high risk regional, rural and remote Australian cohort and the potential impact of alternate pre-analytical protocols aimed at minimising glycolysis. Poor stabilisation of glucose in OGTT samples likely results in significant under-diagnosis of GDM in these settings, where large delays to laboratory analysis are unavoidable. Screening outcomes are inconsistent due to wide variation in analytical delay, and a simple linear adjustment of OGTT results by clinicians is not suitable. Due to the small window available for testing and intervention, accurate diagnosis of GDM is crucial to avert adverse outcomes. Pre-analytical standardisation of glycolysis can be achieved by use of FC Mix tubes. The benefits of improved GDM detection and management in high-risk women from regional, rural, and remote settings should be considered when debating the use of these tubes. Validation of new fasting plasma glucose diagnostic criteria for use with FC Mix collections is warranted to avert unnecessary increased burden for GDM management.

## Figures and Tables

**Figure 1 ijerph-16-04488-f001:**
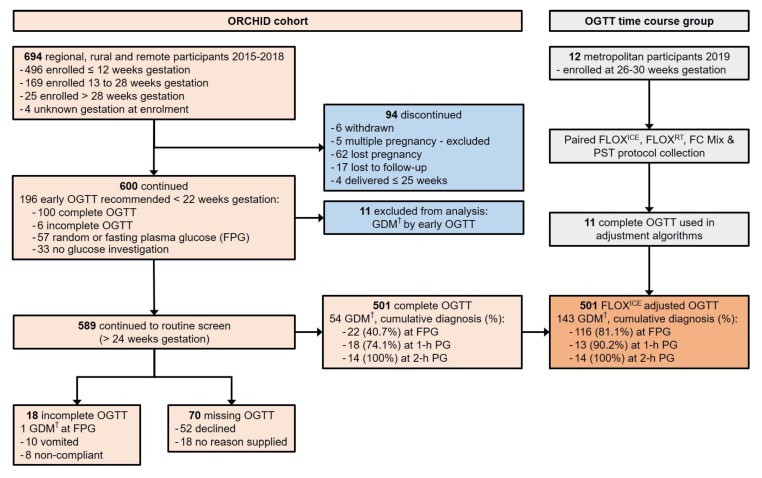
Flowchart for prospective ORCHID cohort participation and OGTT time course group participation, documenting completion of recommended screening for hyperglycaemia in pregnancy by 75 g oral glucose tolerance test (OGTT). ORCHID = Optimisation of Rural Clinical and Haematological Indicators for Diabetes in pregnancy; GDM = gestational diabetes mellitus; PG = plasma glucose; FLOX^ICE^ protocol = fluoride-oxalate (FLOX) samples stored immediately on ice; FLOX^RT^ protocol = FLOX samples stored at room temperature (RT); FC Mix protocol = fluoride-citrate-EDTA samples stored at RT; PST protocol = lithium-heparin plasma separation tubes centrifuged within 5 min of collection at 1300× *g* for 5 min and stored at RT. ^†^ GDM diagnosed if one or more International Association for Diabetes in Pregnancy Study Groups (IADPSG) 2010 glucose criteria were met: FPG ≥ 5.1 mmol/L; 1-h PG ≥ 10.0 mmol/L; 2-h PG ≥ 8.5 mmol/L.

**Figure 2 ijerph-16-04488-f002:**
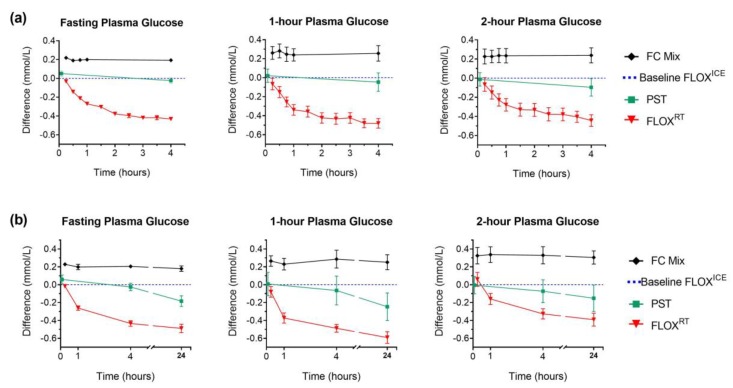
Plasma glucose (PG) in different preservatives compared to FLOX^ICE^ protocol: fluoride-oxalate (FLOX) samples stored immediately on ice; mean glucose concentration measured in plasma from whole-blood aliquots centrifuged at 15 min, 30 min, and 45 min were used to calculate baseline FLOX^ICE^ value (dotted blue line). Lithium-heparin plasma separation tubes (PST) were centrifuged within 5 min of collection; separated plasma fraction remained in the collection tube for storage and measurement. All comparison samples; fluoride-citrate-EDTA (FC Mix); centrifuged PST; and fluoride-oxalate (FLOX^RT^), were held at room temperature for the first 4 h post-collection (fasting PG, *n* = 12; 1-h and 2-h PG, *n* = 11). Plasma for PG measurement separated from FC Mix comparison samples following centrifugation of whole-blood aliquots at regular time intervals (h: 0.25; 0.5; 0.75; 1; and 4). Plasma for PG measurement separated from FLOX^RT^ samples following centrifugation of whole-blood aliquots collected at regular time-intervals (h: 0.25; 0.5; 0.75; 1; 1.5; 2; 2.5; 3; 3.5; and 4). For FC Mix and FLOX^RT^ samples, time represents the time from collection to centrifugation. For PST samples time represents time from collection to centrifugation for initial measurement (5 min) and time from collection to analysis for subsequent measures. (**a**) Data represent mean (± SEM) difference in PG over 4 h from each comparison sample compared to baseline PG (fasting PG, *n* = 12; 1-h and 2-h PG, *n* = 11). (**b**) Following 4 h incubation at room temperature, comparison samples from six participants were transferred to refrigerator (4 °C) for overnight storage. At 24 h, plasma was separated from FC Mix and FLOX^RT^ samples following centrifugation of whole-blood aliquots for PG measurement; plasma fraction from previously centrifuged PST remained in the tube for overnight storage and PG measurement. Data represent mean (± SEM) difference in PG from each comparison sample at 24 h compared to baseline PG (*n* = 6). Difference in PG for FC Mix and FLOX^RT^ samples at 0.25 h, 1 h, and 4 h compared to baseline and PST comparison samples at 5 min, 1 h, and 4 h compared to baseline, plotted for the six participants for direct comparison with 24 h difference.

**Table 1 ijerph-16-04488-t001:** Maternal and newborn characteristics of prospective ORCHID study cohort participants eligible for screening after 24 weeks, stratified by completion of OGTT.

Characteristic	Total Cohort *n* (%)	OGTT Complete *n* = 501 (85.1%)	OGTT not Done or Incomplete *n* = 88 (14.9%)	*p*-Value
**Maternal characteristics**				
Age (years)	589 (100%)	28.8 ± 5.7	27.1 ± 5.0	0.008
Ethnicity				< 0.001
Non-Indigenous	324 (55.0%)	300 (92.6%)	24 (7.4%)	
Aboriginal and Torres Strait Islander	229 (38.9%)	169 (73.8%)	60 (26.2%)	
Other high-risk ethnicity ^†^	36 (6.1%)	32 (88.9%)	4 (11.1%)	
BMI after 24 weeks (kg/m^2^)	589 (100%)	28.5 ± 6.1	28.5 ± 7.2	0.92
BMI category ^‡^				0.28
Normal or underweight (≤ 28.4)	330 (56.0%)	281 (85.2%)	49 (14.8%)	
Overweight (28.5–32.9)	130 (22.1%)	115 (88.5%)	15 (11.5%)	
Obese (≥ 33.0)	129 (21.9%)	105 (81.4%)	24 (18.6%)	
Parity (prior delivery ≥ 20 weeks) ≥ 1 at enrolment	401 (68.1%)	336 (83.8%)	65 (16.2%)	0.21
Family history of diabetes ^§^	181 (30.7%)	146 (80.7%)	35 (19.3%)	0.046
Any antenatal smoking	162 (27.5%)	116 (71.6%)	46 (28.4%)	< 0.001
Antenatal urinary tract infection	30 (5.1%)	26 (86.7%)	4 (13.3%)	0.80
Previous caesarean delivery ^¶^	70 (17.5%)	62 (88.6%)	8 (11.4%)	0.23
Length of gestation at first antenatal presentation (weeks)	589 (100%)	9.8 ± 5.8	11.2 ± 7.1	0.057
Remoteness classification of health service				0.220
MMM2	50 (8.5%)	48 (96.0%)	2 (4.0%)	
MMM3	383 (65.0%)	320 (83.6%)	63 (16.4%)	
MMM6	78 (13.2%)	72 (92.3%)	6 (7.7%)	
MMM7	78 (13.2%)	61 (78.2%)	17 (21.8%)	
**Newborn characteristics**				
Gestational age at delivery (weeks)	589 (100%)	39.2 ± 1.5	38.8 ± 1.6	0.015
Birthweight (g)	589 (100%)	3458 ± 531	3221 ± 513	< 0.001
Length (cm) ^#^	586 (99.4%)	50.6 ± 2.7	49.3 ± 2.5	< 0.001
Head circumference (cm) ^††^	587 (99.7%)	34.6 ± 1.6	34.2 ± 1.5	0.025
Sex—Male	303 (51.4%)	266 (87.8%)	37 (12.2%)	0.056

ORCHID = Optimisation of Rural Clinical and Haematological Indicators for Diabetes in pregnancy study; OGTT = 75 g oral glucose tolerance test; BMI = body mass index; MMM = Modified Monash Model. Data include women eligible for screening after 24 weeks gestation; Data are mean ± standard deviation for continuous variables. For categorical variables, data are number (%) of cohort for total cohort or number (%) of total with characteristic for OGTT group. Two-sided *t*-test *p*-value reported for comparison between groups for continuous data. Pearson Chi-square test *p*-value reported for comparison between groups for categorical data, with the exception of differences in remoteness classification, which was compared by Cuzick non-parametric test for trend across ordered groups. ^†^ Other high-risk ethnicity includes African (*n* = 3), Asian (*n* = 8), Indian (*n* = 4), Middle-Eastern (*n* = 3), Maori (*n* = 13), and Pacific-Islander (*n* = 5) women. ^‡^ BMI, maternal weight in kilograms divided by the square of maternal height in meters, calculated after 24 weeks gestation. BMI categorised according to World Health Organization criteria as adjusted by Hyperglycaemia and Pregnancy Outcomes Study Group to account for maternal weight gain by 24 weeks gestation. ^§^ Family history of diabetes includes women with a first-degree relative with type 1 or type 2 diabetes or history of GDM. ^¶^ Previous caesarean delivery includes only women of parity (prior delivery ≥ 20 weeks) ≥ 1 at enrolment. ^#^ Length data only available for 498 participants with complete OGTT. ^††^ Head circumference only available for 499 participants with complete OGTT.

**Table 2 ijerph-16-04488-t002:** Unadjusted and FLOX^ICE^ algorithm adjusted plasma glucose (mmol/L) measured by standard laboratory procedures compared to FLOX^ICE^ protocol in ORCHID time-course participants.

OGTT Sample	OGTT by FLOX^ICE^ Protocol	OGTT by Standard Procedures			Standard OGTT Adjusted by FLOX^ICE^ Algorithm	
	PG	Delay to Analysis (h)	PG	PG Difference v FLOX^ICE^ Protocol	PG	PG Difference v FLOX^ICE^ Protocol
Fasting	4.9 ± 0.47	3.0 ± 0.4	4.5 ± 0.51 ***	−0.42 [−0.47 to −0.37] (−9.9% to −7.5%)	4.9 ± 0.50	0.02 [−0.04 to 0.07] (−0.7% to 1.4%)
1-h	6.5 ± 1.76	2.0 ± 0.4	6.1 ± 1.67 ***	−0.38 [−0.47 to −0.28] (−6.8% to −4.8%)	6.5 ± 1.64	−0.01 [−0.12 to 0.10] (−1.1% to 1.8%)
2-h	5.8 ± 1.25	1.1 ± 0.4	5.6 ± 1.12 **	−0.26 [−0.42 to −0.10] (−7.0% to −1.5%)	5.8 ± 1.07	−0.02 [−0.18 to 0.13] (−2.9% to 3.2%)

FLOX^ICE^ = Fluoride-oxalate (FLOX) samples stored immediately on ice; ORCHID = Optimisation of Rural Clinical and Haematological Indicators for Diabetes in pregnancy study; PG = plasma glucose; OGTT = 75 g oral glucose tolerance test. Data include 12 participants with paired FLOX^ICE^ samples collected at the same time as their standard OGTT (1-h and 2-h samples, *n* = 11). Data are mean (± SD) for delay to analysis (h) and PG (mmol/L) and mean (95% CI: absolute (mmol/L) and proportional (%)) for difference in PG compared to samples processed by FLOX^ICE^ protocol. PG by standard procedures, unadjusted and algorithm adjusted, compared to FLOX^ICE^ protocol using students paired *t*-test. *p*-value < 0.01 **, < 0.001 ***. FLOX^ICE^ protocol: FLOX samples stored immediately on ice. Three aliquots, taken at 15 min, 30 min, and 45 min (< 1 h post-collection) were centrifuged for separation and immediate PG measurement. FLOX^ICE^ PG data is the mean glucose over the three < 1 h post-collection sample aliquots. OGTT by standard procedure for site: samples collected into FLOX tubes and stored and transported to laboratory at room temperature. Time of analysis of standard procedure samples was recorded and delay to analysis calculated from time of collection. Standard OGTT results were adjusted by linear regression β-coefficients for PG + delay-to-analysis + delay-to-analysis^2^ + constant, estimated for FLOX^ICE^ protocol PG using FLOX^RT^ time-course data (Table A2).

**Table 3 ijerph-16-04488-t003:** Unadjusted and adjusted FC Mix algorithm plasma glucose (mmol/L) measured by standard laboratory procedures compared to FC Mix protocol in ORCHID time-course participants.

OGTT Sample	OGTT by FC Mix Protocol	OGTT by Standard Procedures			Standard OGTT Adjusted by FC Mix Algorithm	
	PG	Delay to Analysis (h)	PG	PG Difference v FC Mix Protocol	PG	PG Difference vs. FC Mix Protocol
Fasting	5.1 ± 0.47	3.0 ± 0.4	4.5 ± 0.51***	−0.62 [−0.67 to −0.56] (−13.6% to −10.9%)	5.1 ± 0.50	−0.01 [−0.06 to 0.05] (−1.3% to 1.0%)
1-h	6.8 ± 1.88	2.0 ± 0.4	6.1 ± 1.67***	−0.63 [−0.84 to −0.43] (−11.3% to −7.1%)	6.8 ± 1.74	−0.01 [−0.21 to 0.18] (−2.2% to 2.8%)
2-h	6.0 ± 1.28	1.1 ± 0.4	5.6 ± 1.12**	−0.49 [−0.73 to −0.26] (−10.8% to 4.2%)	6.0 ± 1.28	−0.03 [−0.20 to 0.14] (−3.1% to 3.5%)

FC Mix = Fluoride-citrate-EDTA samples stored at room temperature; ORCHID = Optimisation of Rural Clinical and Haematological Indicators for Diabetes in pregnancy study; PG = plasma glucose; OGTT = 75 g oral glucose tolerance test. Data include 12 participants with paired FC Mix samples collected at the same time as their standard OGTT (1-h and 2-h samples, *n* = 11). Data are mean (± SD) for delay to analysis (h) and PG (mmol/L) and mean (95% CI: absolute (mmol/L) and proportional (%)) for difference in PG compared to samples processed by FC Mix protocol. PG by standard procedures compared to FC Mix protocol using students paired *t*-test. *p*-value < 0.01 **, < 0.001 ***. FC Mix protocol: FC Mix samples stored at room temperature. Five aliquots taken at 0.25 h, 0.5 h, 0.75 h, 1 h, and 4 h post-collection were centrifuged for plasma separation and immediate PG measurement. FC Mix PG data is the mean glucose over the five sample aliquots. OGTT by standard procedure for site: samples collected into fluoride-oxalate tubes and stored and transported to laboratory at room temperature. Time of analysis of standard procedure samples was recorded and delay to analysis calculated from time of collection. Standard OGTT results were adjusted by linear regression β-coefficients for PG + delay-to-analysis + delay-to-analysis^2^ + constant estimated for FC Mix protocol PG using FLOX ^RT^ time-course data (Table A2).

**Table 4 ijerph-16-04488-t004:** Unadjusted and adjusted OGTT, by FLOX^ICE^, or FC Mix algorithm, in ORCHID prospective cohort participants and proportion with GDM, stratified by ethnicity.

	Mean Glucose ± SD (mmol/L)			*n* (%) Above Diagnostic Threshold at Each OGTT Sample			Cumulative *n* (%) Diagnosed with GDM		
	Aboriginal (*n* = 169)	Other Ethnicity (*n* = 332)	*p*-Value	Aboriginal (*n* = 169)	Other Ethnicity (*n* = 332)	*p*-Value	Aboriginal (*n* = 169)	Other Ethnicity (*n* = 332)	*p*-Value
**Unadjusted OGTT**									
Fasting PG	4.2 ± 0.49	4.3 ± 0.40	0.005	9 (5.3%)	13 (3.9%)	0.467	9 (5.3%)	13 (3.9%)	0.467
1-h PG	7.1 ± 1.74	6.9 ± 1.72	0.208	10 (5.9%)	18 (5.4%)	0.819	14 (8.3%)	26 (7.8%)	0.860
2-h PG	6.1 ± 1.63	5.9 ± 1.39	0.245	13 (7.7%)	18 (5.4%)	0.319	20 (11.8%)	34 (10.2%)	0.587
**FLOX^ICE^ adjusted OGTT ^†^**									
Fasting PG	4.7 ± 0.47	4.8 ± 0.41	0.022	30 (17.8%)	86 (25.9%)	0.041	30 (17.8%)	86 (25.9%)	0.041
1-h PG	7.6 ± 1.67	7.4 ± 1.69	0.114	14 (8.3%)	23 (6.9%)	0.583	33 (19.5%)	96 (28.9%)	0.023
2-h PG	6.4 ± 1.52	6.2 ± 1.32	0.126	16 (9.5%)	25 (7.5%)	0.454	39 (23.1%)	104 (31.3%)	0.053
**FC Mix adjusted OGTT**									
Fasting PG	4.9 ± 0.46	5.0 ± 0.41	0.017	49 (29.0%)	147 (44.3%)	0.001	49 (29.0%)	147 (44.3%)	0.001
1-h PG	8.0 ± 1.77	7.7 ± 1.80	0.110	17 (10.1%)	37 (11.1%)	0.711	54 (32.0%)	159 (47.9%)	0.001
2-h PG	6.8 ± 1.76	6.5 ± 1.52	0.111	24 (14.2%)	32 (9.6%)	0.125	61 (36.1%)	166 (50.0%)	0.003

ORCHID = Optimisation of Rural Clinical and Haematological Indicators for Diabetes in pregnancy study; OGTT = 75 g oral glucose tolerance test; PG = plasma glucose; FLOX^ICE^ = fluoride-oxalate (FLOX) samples stored immediately on ice; FC Mix = fluoride-citrate-EDTA samples stored at room temperature. Data include 501 participants with complete OGTT women eligible for screening after 24 weeks gestation; Data are mean glucose ± SD (mmol/L) for ethnic group or number (%) of ethnic group with OGTT outcome. Two-sided *t*-test *p*-value reported for comparison between ethnic group for continuous data. Pearson Chi-square test *p*-value reported for comparison between ethnic group for categorical data. OGTT results were adjusted by linear regression β-coefficients for PG + delay-to-analysis + delay-to-analysis^2^ + constant for estimated FLOX^ICE^ protocol PG or FC Mix protocol PG from ORCHID time-course participants.

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
