# Peer review of "Real-World Gestational Diabetes Screening: Problems with the Oral Glucose Tolerance Test in Rural and Remote Australia"

_ijerph, 2019, doi:10.3390/ijerph16224488_

Round 1
Reviewer 1 Report
Although this reviewer warmly welcomes this manuscript, some questions should be addressed:
The rationale behind performing the study is not clearly presented to the scientific audience. A more integrated appraisal of the relevant literature would be appropriate to provide the context for the study.
The study strengths and limitations should be deeply addressed, taking into account sources of potential bias or imprecision: Discuss both direction and magnitude of any potential bias.
Please carefully revise wording and abbreviation throughout the manuscript.
Reviewer 2 Report
International Journal of Environmental Research and Public Health
Title: Real-world gestational diabetes screening: problems with the oral glucose tolerance test in rural and remote Australia
Summary: The authors showed that the measurement of blood glucose levels after OGTT is variable, depending on the methods used for sample storing and time to analysis. It can lead to lower blood glucose levels than genuine, leading to an underdiagnosis of diabetes. The authors tested different methods for sample storing and handling and provide a practical improved alternative.
Broad comments: The study is well done. The number of probands is high enough to obtain significant results. Many parameters are collected, analysed and presented. The results regarding the blood glucose levels are nice and convincing and the proposed change to other tubes is reasonable.
What should also be discussed and eventually analysed is the duration of fasting. The differences in fasting blood glucose are the most meaningful according to the paper, but the recommended time of fasting in the different facilities is quite variable and it is not written what the duration really is. There is maybe a difference in fasting between probands from certain centres and some ethnicities could be located preferentially to some centres.
Specific comments:
Line 33: "the International Diabetes in Pregnancy Study Group (IADPSG)" it is "International Association of the Diabetes and Pregnancy Study Groups".
Line 176 Table 1: difficult to read, could maybe improved.
Line 211 - : " All comparison samples; fluoride-citrate-EDTA (FC Mix); heparinised plasma separation tube (PST); and fluoride-oxalate (FLOXRT), were held at room temperature for the first 4-hours post-collection (n=12). PST were centrifuged within 5-minutes of collection"
PST stored 4h RT or centrifuged within 5 minutes? Where are the 15 min, 30 min...values coming from?
Line 217 - : "(b) Following 4-hour incubation at room-temperature, comparison samples from six participants were transferred to refrigerator (4C) for overnight storage. At 24-hours, Data represent mean (± SEM) difference in PG from each comparison sample over 24-hours compared to baseline PG (n=6).
In Fig 2b are also the values of 1h and 4 h.
Please improve legend of figure
Line 342 and 343: please insert "Ref number" of Song et al. and there is no "." after et
Round 2
Reviewer 1 Report
No further comments
